# Mitostasis, Calcium and Free Radicals in Health, Aging and Neurodegeneration

**DOI:** 10.3390/biom11071012

**Published:** 2021-07-10

**Authors:** Juan A. Godoy, Juvenal A. Rios, Pol Picón-Pagès, Víctor Herrera-Fernández, Bronte Swaby, Giulia Crepin, Rubén Vicente, Jose M. Fernández-Fernández, Francisco J. Muñoz

**Affiliations:** 1Laboratory of Molecular Physiology, Departament de Ciències Experimentals i de la Salut, Universitat Pompeu Fabra, 08003 Barcelona, Spain; jgodoz@gmail.com (J.A.G.); pol.picon@upf.edu (P.P.-P.); victor.herrera@upf.edu (V.H.-F.); bronte.swaby@gmail.com (B.S.); Giuliacrepin96@gmail.com (G.C.); ruben.vicente@upf.edu (R.V.); jmanuel.fernandez@upf.edu (J.M.F.-F.); 2Escuela de Medicina, Facultad de Medicina y Ciencia, Universidad San Sebastián, Santiago 7510157, Chile; Juvenal.rios@umayor.cl; 3Programas para el Futuro, Facultad de Estudios Interdisciplinarios y Facultad de Ciencias, Universidad Mayor, Santiago 8580745, Chile

**Keywords:** mitochondria, mitostasis, calcium, oxidative stress, nitric oxide, aging

## Abstract

Mitochondria play key roles in ATP supply, calcium homeostasis, redox balance control and apoptosis, which in neurons are fundamental for neurotransmission and to allow synaptic plasticity. Their functional integrity is maintained by mitostasis, a process that involves mitochondrial transport, anchoring, fusion and fission processes regulated by different signaling pathways but mainly by the peroxisome proliferator-activated receptor-γ coactivator-1α (PGC-1α). PGC-1α also favors Ca^2+^ homeostasis, reduces oxidative stress, modulates inflammatory processes and mobilizes mitochondria to where they are needed. To achieve their functions, mitochondria are tightly connected to the endoplasmic reticulum (ER) through specialized structures of the ER termed mitochondria-associated membranes (MAMs), which facilitate the communication between these two organelles mainly to aim Ca^2+^ buffering. Alterations in mitochondrial activity enhance reactive oxygen species (ROS) production, disturbing the physiological metabolism and causing cell damage. Furthermore, cytosolic Ca^2+^ overload results in an increase in mitochondrial Ca^2+^, resulting in mitochondrial dysfunction and the induction of mitochondrial permeability transition pore (mPTP) opening, leading to mitochondrial swelling and cell death through apoptosis as demonstrated in several neuropathologies. In summary, mitochondrial homeostasis is critical to maintain neuronal function; in fact, their regulation aims to improve neuronal viability and to protect against aging and neurodegenerative diseases.

## 1. Introduction

The brain is the most energy-demanding organ in the body, consuming up to 20% of the total oxygen supply and up to 20% of total glucose intake [1], especially in the neuropil [2]. This energy is needed for normal cellular functions, but in particular, neurons must restore ionic gradients to allow action potentials, carry out neurotransmission, provide the machinery for synaptic plasticity, and maintain healthy and functional membranes along axons and dendrites.

Adenosine triphosphate (ATP) is mainly produced by glucose metabolism and through the mitochondrial respiratory chain (MRC). Therefore, mitochondria need to be functionally active in soma, axons and dendrites throughout the life of an individual. Axonal transport of mitochondria, and mitochondrial fission and fusion, contribute tomaintaining a functional and healthy pool of mitochondria. These mitochondria are relocated continuously along the neurons depending on the local energy requirements [3]. The complex, dynamic system that maintains and delivers mitochondria is termed mitostasis and it regulates the number, trafficking and protein turnover of mitochondria in each compartment of the neuron [4].

On the other hand, mitochondria participateactively in calcium (Ca^2+^) homeostasis, a process reviewed in the present work, and this mitochondrial function is fundamental for neuronal activity [5].To achieve this aim, they are physically and functionally linked to the endoplasmic reticulum (ER) by specialized structures, mitochondria-associated membranes (MAMs), which have been proposed to be highly relevant in calcium buffering. They are sites of juxtaposed ER and outer mitochondrial membrane (OMM), which facilitate organelle communication, allowing the regulation of Ca^2+^ fluxes from ER to mitochondria, among other cellular functions such aslipid synthesis and transport, and glucose metabolism [6]. Ca^2+^ also regulates mitochondrial functions and itsdyshomeostasisaffects the MRC, triggering free radical production [7]. Reactive oxygen species (ROS) production, Ca^2+^ uptake and mitochondrial membrane depolarization have been linked to neuronal apoptosis [8]. They disrupt the normal mitochondrial functioningthrough the uncoupling of MRC, compromising energy production [7].

Considering the long life of the neurons, mitostasis is fundamental to maintain healthy and functional neurons even during aging. Here, we present a review of the role of the factors implicated in mitostasis and its relationship with Ca^2+^ and free radicals in health, aging and neurodegenerative processes. 

## 2. Mitostasis in Neurons

The maintenance and distribution of the mitochondrial reserve is termed mitostasis, which includes mitochondrialtransport, anchoring, fission and fusion. In fact, most of the mitochondria do not exist as small, distinct organelles, but rather form a highly interconnected reticulum with contiguous membranes [9]. This reticulum is highly dynamic with ongoing divisions and reconnections. The balance of fission and fusion determines the length of mitochondria, and it is regulated by stress and nutrient availability [10]. Mitostasis also involves protein reparation or degradation, and the formation of mitochondria-derived vesicles yielding to mitophagy and macroautophagy [4].

In particular, mitochondrial biogenesis requires materials from the cytoplasm and nucleus, since mammalian mitochondria DNA (mtDNA) encodes only 13 proteins [11]. For this reason, mitochondrial biogenesis requires the importation of over 1500 proteins encoded by the nuclear genome [12] and phospholipids for the inner mitochondrial membrane (IMM) and the OMM [13].

### 2.1. Mitochondrial Trafficking

The location of mitochondria, depending on local energy demands, is critical for polarized cells such as neurons. Interconnected mitochondria are located in neuronal cell soma. However, to enter the axon, a mitochondrion must undergo a fission reaction. This reaction frees it from its mitochondrial reticulum.

Once released, mitochondrial fast anterograde transport is mediated by microtubules [14]. Mitochondrial Rho GTPase 1 and 2 (Miro 1 and 2) serve as Ca^2+^ sensors that regulate kinesin-mediated mitochondrial motility. Miro 1 and 2 have an outer C-terminal transmembrane domain with GTPase domains, which interact with the trafficking kinesin protein (TRAK)/Milton family of proteins [15,16,17,18,19]. These cargo adaptorsbind to Kinesin-1 anddynein/dynactin, beingcritical in mitochondrial trafficking and its spatial distribution [20]. Kinesin-1 cargoes, often together with dynein, drive mitochondria transport into the axon [18,21], and the dynein/dynactin cargo complextransports mitochondria into dendrites [20,22]. 

On the other hand, shorter-range mitochondrial transport is actin-dependent anduses the dynein/dynactin complex as actin-mitochondrial anchorage. The cargo Myosin 19 (Myo19) [23] has been suggested as critical for this process [24].

### 2.2. Mitochondrial Anchoring

Mitochondrial anchoring to inner cell structures is needed to provide the required energy at the right place, as around two-thirds of neuronal mitochondria remain stationary [25]. These pools of mitochondria are mainly located in the soma and in areas with high ATP demand, such as the nodes of Ranvier, axonal presynaptic endings and dendrites [26]. Mitochondria anchoring mechanisms are not fully understood and they seem to involve different proteins depending on the localization of the mitochondria. For instance, in axonal presynaptic endings, activation of adenosine monophosphate-activated protein kinase (AMPK) by synaptic activity leads to myosin VI phosphorylation, ending with mitochondrial recruitment and syntaphilin-mediated anchoring on presynaptic filamentous actin [27]. This process is specific to the stationary mitochondria, not being related to the mitochondria mobility machinery [3,28]. Remarkably, it is an axon exclusive physiological mechanism. On the other hand, axonal mitochondrial anchor mislocalization to dendrites has been associated with multiple sclerosis disease [29].

### 2.3. Mitochondrial Fission

Mitochondrial fission is essential for shaping the mitochondrial network and is required for generating additional mitochondria of the appropriate size for their transport by molecular motors along the cytoskeleton [30] (Figure 1). 

Mitochondria fission starts when the mitochondrial fission factor (MFF) surrounds the OMM forming a ring [31,32]. Mitochondrial fission 1 (Fis1) and mitochondrial dynamics proteins of 49 and 51 kDa (MID49 and MID51) recruit the cytoplasmic Dynamin-related protein 1 (Drp1) [33,34,35]. Drp1 translocates to the mitochondria and binds to MFF, inducing a conformational change due to its GTPase activity. This conformational change produces the ring contraction that starts mitochondrial fission [33,36]. Cellular cytoskeleton further separates both mitochondria depending on the MAMs. The Inverted Formin 2 protein (INF2) is localized in MAMs and its activation causes actin polymerization, which generates the driven force separating both mitochondria [37].

### 2.4. Mitochondrial Fusion

Mitochondrial fusion is a cell protective mechanism [38] based in two different and concomitant processes, the OMM fusion and the IMM fusion (Figure 2).

The dynamin-associated GTPases Mitofusin proteins 1 and 2 (MFN1 and MFN2) are responsible for the OMM fusion [39,40]. Both GTPases share a high structural and biochemical homology, but they have different roles [41]. MFN1 is found exclusively in the OMM, and allows the contact between the OMM of different mitochondria [42]. MFN2, found in mitochondria and ER, has an anchoring role since it joins the two organelles, forming homotypic and heterotypic complexes with MFN1 [43]. The spiral domains of MFN1 and MFN2 interact, forming homo- or hetero-oligomeric complexes through GTP hydrolysis in the fusion spot [44]. However, there are controversial reports regarding the relevance of this mechanism since it was observed that the loss of MFN-2 increased the ER–mitochondrial juxtaposition [45]. This fact does not contradict that the functionality of mitochondria–ER contacts is affected in MFN2 KO cells, as suggested by the previous observation that Ca^2+^ transfer from the ER to the mitochondria is altered in these cells [43].

The optic atrophy 1 protein (OPA1) mediates the IMM fusion [46,47]. The long form of the protein (l-OPA1) works as a coordinator of the IMM of both fusing mitochondria. The two membranes fuse through the short isoform (s-OPA1) by its GTPase activity [48]. OPA1 is also associated with the maintenance of the MRC by cristae organization, and the control of apoptosis [49,50,51,52], demonstrating versatility of functions. 

### 2.5. The regulator of Mitochondrial Biogenesis PGC-1α

The peroxisome proliferator-activated receptor-γ (PPARγ) coactivator-1 α (PGC-1α) is a member of a transcription coactivator family, composed ofPGC-1α, PGC-1β and PGC-related coactivator (PRC). These proteins play a central role in the regulation of mitochondrial biogenesis and cellular energy metabolism [53] (Figure 3).

The action mechanism of PGC-1α implies its interaction with two nuclear transcription factors, nuclear respiratory factors 1 and 2 (NRF-1 and 2), which activate the mitochondrial transcription factor A (TFAM), a mtDNA-binding protein [54]. TFAM and the cofactor mitochondrial transcription factor B (TFB1M) are essential for mitochondrial gene expression [55]. This molecular pathway promotes the equilibrium in the assembly of some components of the MRC, which ultimately ensures appropriate mitochondrial function, mtDNA replication and apoptosis control [56,57,58].

The expression and/or activity of PGC-1α is regulated by different intracellular signaling pathways such as the PGC-1α regulatory cascade [59], thyroid hormone (TH) [60], sirtuin 1 (SIRT1) [61,62], calcineurin [63], AMPK [64], cyclin-dependent kinases (CDKs) [65], and β-adrenergic signaling [66]. These regulators of PGC-1α aim to coordinate the expression and function of mitochondrial proteins during mitochondrial biogenesis [67]. Additionally, not only transcriptional regulation but also post-translational modifications allow the control of the PGC-1α activity. For example, PGC-1α phosphorylation by p38 mitogen-activated protein kinase alpha (p38α) enhances its activity [68], and PGC-1α acetylation suppresses its activity [61].

PGC-1α and mitochondrial biogenesis play important roles in the formation and maintenance of dendritic spines and synapses in hippocampal neurons [69]. In this context, PGC-1α also controls the synthesis of the Brain-Derived Neurotrophic Factor (BDNF) [70] and the activity of Extracellular Signal-Regulated Kinases (ERKs) [71]. BNDF has been widely reported in the literature as a neuroprotective agent produced by physical exercise [72] and, interestingly, exercise induces the upregulation of PGC-1α expression and increases mitochondrial mass within the brain [73]. In addition to its role in mitochondrial biogenesis, PGC-1α also regulates apoptosis and the ROS detoxification system [74].

## 3. The Role of Mitochondria in Calcium Homeostasis

Mitochondria regulate the intracellular Ca^2+^ in tight association with the ER, but mitochondrial Ca^2+^ also controls mitostasis and even neuronal fate (Figure 4).

### 3.1. Calcium Entrance to Mitochondria

Ca^2+^ entry from the cytosol into the mitochondria requires its flow through both the OMM and the IMM. The first step is carried out by the voltage-dependent anion channel 1 (VDAC1), the most abundant protein at the OMM. VDAC1 is a channel that allows the entry and exit of multiple metabolites and ions. It shows two voltage-dependent conductance states [75]. At the high conductance state (at low potentials of −20 to +20 mV), it is permeable even to large metabolites and cations, such as acetylcholine, dopamine, glutamate and ATP [76,77]. At the low conductance state, at higher positive or negative potentials (>30–60 mV), the selective permeability shifts to small cations, including Ca^2+^ [78,79,80].

The transport of Ca^2+^ into the mitochondrial matrix from the intermembrane space is mainly mediated by the membrane Ca^2+^ uniporter (MCU). This protein constitutes a heteromeric channel complex at the IMM along with mitochondrial calcium uniporter dominant negative subunit beta (MCUb), which is a paralog of MCU acting as dominant negative subunit [81]. Essential MCU regulator (EMRE), mitochondrial calcium uptake 1 and 2 (MICU1,2) and MCU regulator 1 (MCUR1) have multiple functions as scaffold factors for the MCU complex function, promoting mitochondrial bioenergetics and the activation of the solute carrier 25A23 (SLC25A23) [82,83,84,85]. The efficiency of Ca^2+^ influx through this channel complex in the mitochondria depends on the highly negative mitochondrial membrane potential (ΔΨm), produced by the pumping of protons across the IMM into the intermembrane space by complexes I, III, and IV of the MRC. Such ΔΨm allows ATP synthesis bycomplex V (ATP synthase) and determines the electrochemical driving force for Ca^2+^ movement. MCU is the pore-forming subunit, whose activity is regulated by MCUb (a paralog of MCU acting as dominant negative subunit) [81], MICU1 and MICU2 (as Ca^2+^-dependent modulators), and MCUR1 and SLC25A23 (as positive regulators). EMRE has been shown to be essential for MCU-mediated mitochondrial Ca^2+^ uptake [86]. Thus, EMRE is considered as the scaffolding protein-mediating interaction of MICU1 and MICU2 with MCU [87]. Through their EF-hand Ca^2+^-binding motifs, MICU1 and MICU2 sense the Ca^2+^ at the intermembrane space to modulate pore gating [88,89]. At resting low Ca^2+^ levels, they inhibit MCU channel activity [90,91,92]. Once Ca^2+^ rises above 1–2 μM, MICU1 and MICU2 drive channel gating, with MICU1 working as a positive regulator, which promotes the MCU open conformation [88,93,94], and MICU2 slowing channel activation. It results in the limitation of the mitochondrial Ca^2+^ uptake for the proper control of Ca^2+^ homeostasis [92,94]. Besides, MICU1 and MICU2 have been reported to form heterodimers that respond to sub-micromolar Ca^2+^ in a steep co-operativity manner, thus defining MCU threshold and functioning as ON/OFF switches for mitochondrial Ca^2+^ uptake [90]. The enhancement of MCU channel activity induced by MCUR1 seems to be related with its role as a scaffold factor required for the correct assembly of the whole uniporter complex [95].

Steady-state levels of mitochondrial Ca^2+^ depend on the dynamic balance between this MCU-mediated Ca^2+^ influx and Ca^2+^ efflux through the Li^+^-permeable Na^+^/Ca^2+^ exchanger (NCLX), also located at the IMM [96].

In this way, the mitochondrial network regulates the local Ca^2+^ concentrations at cellular microdomains where Ca^2+^ signaling is intense, in particular, at the close vicinity of the ER and the plasma membrane. In the plasma membrane they act as intracellular Ca^2+^ buffers, since they are located in the proximity of Ca^2+^ channels. In the ER or the sarcoendoplasmic reticulum, mitochondria regulate the diffusion of Ca^2+^ waves through the cytosol [97]. 

Due to the essential role of Ca^2+^ in cellular homeostasis, its flow is tightly regulated. This process is cell specific through the differential expression of multiple Ca^2+^ channels and pumps both at the plasma membrane and intracellular organelles, such as the ER. The Ca^2+^-buffering function of mitochondria in cytosolic signaling microdomains strongly controls the regulation of many of these Ca^2+^ pathways, because their activity is allosterically modulated by cytosolic Ca^2+^. This is the case of the store-operated Ca^2+^ channel (SOC) [98], ryanodine receptors (RyR) [99], L-type Ca^2+^ channels [100] or transient receptor potential (TRP) channels [101].

### 3.2. Calcium Functions in Mitochondria

Inside the mitochondria, Ca^2+^ plays a crucial role by allowing the regulation of the metabolic energy supply. It works as an indirect enhancer of three proton pumps in the MRC: complexes I, III, and IV [102,103,104]. Besides, Ca^2+^ also upregulates the ATP synthase itself [105,106]. Moreover, mitochondrial Ca^2+^ has been shown to inhibit mitochondrial elimination of reactive oxygen species (ROS) in a dose-dependent manner from 1 to 100 µM. This concentration range makes this mechanism relevant in both physiology and pathology [107]. 

Furthermore, calcium-dependent activation of CAMKIIβ is linked to mitochondrial biogenesis in neurons. This protein is able to activate AMPK, which is one of the main regulators of PGC1α [108].

Finally, both cytosolic Ca^2+^ overload or the imbalance between mitochondrial Ca^2+^ uptake and removal can lead to the toxic accumulation of Ca^2+^ in mitochondria. This often results in the opening of the mitochondrial permeability transition pore (mPTP), which in turn leads to mitochondrial swelling and dysfunction, and finally to cell death through apoptosis or necrosis in several neuropathologies [109,110,111,112,113,114,115], such asAlzheimer’s disease (AD) and hypoxic-ischemic brain injury.

### 3.3. The Role of the MAMs

Mitochondria do not function as isolated structures; in fact, up to a 5–20% of the mitochondrial surface is likely in close contact with the ER [116,117]. These sites are physical/functional platformsinvolved in different fundamental cellular processes. The ER–mitochondrial contacts do not occur randomly, but on specific domains termed MAMs, which have a well-defined structure with several proteins present at both the OMM and the ER membrane [118]. MAMs allow a better transport of Ca^2+^ to mitochondria and regulate spikes of activity through Ca^2+^ transport from the ER to the mitochondria [119].The alteration of MAMs yields to the lack of Ca^2+^ homeostasis and mitochondrial biogenesis impairment [97,116,120,121].

The core of the MAMs is a conserved multiprotein complex formed by the IP3Rs (ER), which interacts with the cytosolic fraction of the mitochondrial chaperone glucose-regulated protein 75 (Grp75) and the VDAC1 [122]. It works as an anchor but also allows Ca^2+^ transfer and favors phospholipid exchange between the two organelles. Other proteins have been proposed to play a key role in MAMs formation and function such as MFN2, an ER–mitochondria anchor [43,123]. The vesicle-associated membrane protein-B (VAPB) and the tyrosine phosphatase-interacting protein-51 (PTPIP51), located at the ER and mitochondria, respectively, have also been described as scaffold-forming dimers [124,125].

In the same way, the phosphofurin acidic cluster sorting protein 2 (PACS-2), which seems to constitute part of the MAMs, also acts as another regulatory signaling protein [126]. Even while the role of PACS-2 as a ER–mitochondria tether scaffold lacks the necessary evidence, the depletion of this protein causes mitochondrial disconnection from the ER [126,127]. It has been proposed that two distinct domains are present in MAMs that differ in their lipid content, function and control of glucose homeostasis [128,129,130,131].

Additionally, it must be notedthat the proteolytic processing of some of the MAMs complexes also act as signaling mechanisms, being able to drive critical processes such as cell death. That is the case of the ER, B-cell receptor-associated protein 31 (BAP31) and Fis1 complex since a sustained/increased efflux of reticular Ca^2+^ to the mitochondria induces the cleavage of the BAP31 leading to neuronal apoptosis [132].

### 3.4. The Role of Calcium in mPTP Opening

Ca^2+^ plays a crucial role in apoptosis either in the extrinsic pathway as a second messenger or in the intrinsic pathway through mitochondrial-mediated death. In the intrinsic pathway, the high mitochondrial Ca^2+^ load opens the mPTP, located in the IMM. The opening of mPTP causes reduction of the ΔΨm, dysfunction of MRC and the release of the cytochrome C (CytC) [133,134,135,136,137]. CytC leads to the formation of a complex with the apoptosis protease-activating factor-1 (APAF-1), that activates the caspase cascade to finally induce cell death [138]. This process is further enhanced by the production of ROS and yields to a lower production of ATP [139]. Moreover, CytC favors mitophagy and impairs mitostasis [140]. The pathological relevance of this mechanism is shown in several neurodegenerative diseases, since an increase in mPTP activity has been reported in AD [136,137], amyotrophic lateral sclerosis (ALS) [141,142,143], Parkinson’s disease (PD) [144], and Huntington’s disease (HD) [145,146]. On the other hand, the opening of the mPTP and mitochondrial dysfunction in mitochondrial diseases promote the production of ROS that deteriorate their own repair and function dynamics, causing a vicious circle that aggravates this pathological process [139].

## 4. ROS Production by Mitochondria

Mitochondria produce ROS, mainly superoxide anion (O_2_·^−^) by the MRC, and indirectly hydrogen peroxide (H_2_O_2_) and hydroxyl radical (OH·^−^) [147] (Figure 5). However, the role of ROS cannot be misunderstood. ROS constitute a cellular signaling system, and neuronal function depends on the subtle control of ROS levels. ROS have a direct influence on the intracellular Ca^2+^ levels since they can modulate the activity of several Ca^2+^ channels including IP3Rs, sarco/endoplasmic reticulum Ca^2+^-ATPase (SERCA) and MCU [148,149,150]. Fine-tuning redox reactions and ion signaling pathways are fundamental for proper neuronal function. Therefore, the antioxidant enzymes superoxide dismutase (SOD) and catalase (CAT), and the glutathione system are important for the maintenance of the normal neuronal redox state and, consequently, neurotransmission and cognitive processes [151,152,153]. In fact, in an aging rat animal model, Cu/Zn SOD (SOD1) overexpression induces an altered redox status and a decrease in N-methyl-D-aspartate receptor (NMDAR) synaptic function, glutathione peroxidase (GPx) activity and glutathione (GSH) levels. SOD1 overexpression also impaired long-term potentiation (LTP) followed by impaired learning. 

On the other hand, aging and neurodegenerative diseases are characterized by an increase in ROS generation, with the consequent alteration of the ΔΨm, Ca^2+^ dyshomeostasis and decreased ATP production. Mitochondrial fission in neurons can be initiated by ROS, where MFN2 plays a protective role, but Drp1-mediated fission increases ROS production [154]. Overall, in vitro experiments demonstrated that cells exposed to ROS have altered morphology of the mitochondrial network, and this morphological alteration depends on the amount of ROS and the duration of the experimental challenge [155]. It has also been demonstrated that the overproduction of ROS can induce further ROS production, generating a permanent pro-oxidative state. This state leads tonucleic acid, protein and lipid oxidation, having a determinant role in cellular fate and neurodegenerative diseases [147].

## 5. Nitric Oxide and Mitochondria

Nitric oxide (NO) has a regulatory effect on mitochondrial function and mitostasis. Mitochondria produce their own NO by a constitutive mitochondrial NO-synthase (mtNOS; an isoform of the neuronal NOS), whose activation depends on Ca^2+^ [156,157,158]. The role of mtNOS has a special significance, because this enzyme produces the vast majority of cellular NO, much more than all the other NOS together [159]. Therefore, mtNOS plays key roles in health and also age-related metabolic alterations [160].

NO favors mitochondrial biogenesis inducing mtDNA replication and PGC-1α, NRF-1 and TFAM expression [161], but it also regulates mitochondrial functions. NO decreases the activity of the MRC through the inhibition of CytC. Thus, mtNOS has a function as a physiological modulator of ATP production [162,163]. NO in high concentrations activates the mPTP [164] leading to apoptosis;however, it could even be protective, since it has been demonstrated that mtNOS has an antiapoptotic role through S-nitrozation of caspase 3, which renders it inactive [165].

On the other hand, NO can react with O_2_·^−^, generating peroxynitrite (ONOO^−^) [166,167]. Peroxynitrite is a highly reactive anion that nitrates proteins and lipids, which negatively impacts their physiological functions, especially in neurodegenerative diseases [156,168,169].

## 6. Mitochondria in Aging and Neurodegeneration

Aging is defined as a progressive time-related accumulation of changes responsible for or at least involved in an increased susceptibility to disease and death. In fact, mitostasis imbalance leads to the development of cognitive decline [170,171] and neurodegeneration [157,172,173].

### 6.1. Physiological Aging 

During aging, neural activity is less localized in certain brain regions, notably in the prefrontal cortex, while performing executive-function related tasks [174,175]. These observations suggest that reductions in higher-order brain functions are accompanied by stereotypical, structural and neurophysiological changes in the brain, with variable degrees of cognitive decline that are significantly induced by aging in the absence of disease [176,177]. This reduction in brain metabolic activity during aging has been proposed to be due to mitochondrial dysfunction, which produces impaired energy supply, imbalanced Ca^2+^ buffering, increased production of ROS, DNA alteration, opening of the mPTP and apoptosis. All these factors plus the dysregulation of different proteins (Table 1) contribute to the progressive decline of long-lived neurons linked to aging, which is exacerbated in neurodegenerative diseases [178].

In particular, mitochondrial dysfunction is an early and common pathological feature of aging in which the activities of complexes I, III, and IV are decreased [179,180,181]. Impairment in mitochondrial function occurs when the activity of complex IV of the MRC is reduced. Simultaneously, complex V, which metabolizes ADP to form ATP, is functionally altered and damaged by oxidation. Therefore, VDAC in the OMM is also damaged by oxidative stress, and these changes contribute to the formation of the mPTP, significantly enhanced in neurodegenerative processes [182,183,184].

According to the mitochondrial theory of aging, the accumulation of mutations in mtDNA due to continuous exposure to ROS is a key factor in cellular aging [185]. DNA alterations are due to the susceptibility of mtDNA to ROS-induced strand breaks and point or deletion mutations. In some cases, the mtDNA repair system fails and mtDNA mutations can be transmitted to and accumulated in daughter mitochondria [186,187].

Regarding Ca^2+^ homeostasis during physiological aging, there is an increase in intracellular Ca^2+^ due to the increased activity of L- and N-type voltage-gated Ca^2+^ channels (VGCC), and the enhanced age-dependent expression of the N-type [188] in the plasmatic membrane. This increase in Ca^2+^ load also contributes to mPTP formation, mitochondrial dysfunction and apoptosis.

**Table 1 biomolecules-11-01012-t001:** Proteins involved in mitochondrial dysfunction in aging.

Molecule	Function	Ref.
DNA polymerase γ (Polγ)	Mitochondrial DNA polymerase	[189]
Hsp70/Hsp90 molecular chaperones	Protection and interaction with protein import receptor Tom70	[190]
Lon Protease	Involved in drug-induced mitochondrial dysfunction and endoplasmic reticulum stress	[191]
Transcription Factor ATF5	Mitochondrial unfolded protein response (mtUPR)	[192]
Estrogen receptor	Involved in mtUPR	[193]
α-ketoglutarate dehydrogenase, and pyruvate dehydrogenase	Mitochondrial respiration	[194]
DRP1	Replication machinery	[195]

### 6.2. Inflammaging

A lowgrade of chronic activation of the inflammatory response associated with aging, termed inflammaging, yields to an inflammatory status that worsens or contributes to the onset of age-associated pathologies [196]. In fact, NF-κB, the master regulator of the inflammatory cascade within cells, increases its activity with aging, being linked to a wide spectrum of age-related disorders [197,198] (Figure 6). The activation of NF-κB promotes the expression of interleukin 1β (IL-1ß) and interleukin 18 (IL-18), two major products of the inflammasome [199,200]. Moreover, NF-κB activation also promotes tumor necrosis factor α (TNF-α) and interleukin 6 (IL-6) expression [200], which contribute to mitochondrial dysfunction [199].

Inflammasomes are multiprotein signaling patterns responsible for the maturation of pro-IL-1β and pro-IL-18. Most inflammasomes consist of an upstream sensor, in most cases an adaptor protein (ASC), and inflammatory caspases such as caspase-1. Upon activation, sensor proteins oligomerize with adaptor proteins, forming large complexes called specks (ASC specks) [201]. There is a close relationship between the inflammasome and mitophagy. Firstly, it acts as a protective mechanism. In fact, the inflammasome induces the expression of PGC-1α [202]. However, once dysregulated it becomes a pathological process that aggravates mitochondrial damage [202] (Table 2).

The inflammasome is modulated by the MAMs [203]. The nucleotide-binding oligomerization domain-like receptor 3 (NOD-like receptor 3 or NLRP3), an inflammasome component, when activated binds to the ER associated with caspase-1 [204,205], promoting its activation, which results in proteolytic activation of IL-1ß and IL-18 [206]. Significantly, NLRP3 is activated by ROS, suggesting inflammasome activation originates in the mitochondria [200,206]. On the other hand, when VDAC1 expression is decreased, the entry of mitochondrial Ca^2+^ is affected, and inflammasome activity is reduced [204]. Accordingly, MAMs, through the exchange of ER and mitochondrial content, including Ca^2+^ and ROS, among others, constitute a highly complex regulatory mechanism able to coordinate several cellular processes, including the inflammatory response [207,208,209].

In fact, the study of pathological conditions with high levels of inflammatory components suggests that age-related disorders such as neurodegenerative diseases and aging itself could be controlled by the modulation of the inflammatory mechanisms [198,210,211].

**Table 2 biomolecules-11-01012-t002:** Proteins involved in mitochondrial dysfunction in imflammaging.

Molecule	Function	Ref.
Cyclic GMP-AMP (cGAS) and the cyclic GMP-AMP receptor stimulator of interferon genes (STING), (cGAS-STING)	Inflammatory response	[212]
Mitochondria-associated viral sensor (MAVS)	Mitochondrial adaptor	[213]
Retinoic acid-inducible gene I (RIG-I)	RNA helicase	[214]
NOD-like receptor family member X1 (NLRX1)	Regulators of MAVS function	[215]
TNF-receptor-associated factor (TRAF)	Type I IFN production	[216]
Succinate dehydrogenase	Inflammatory response	[217]
mtDNA	Dysfunctionby oxidation	[218]
Cardiolipin	Nlrp3 activation	[219]
N-formyl peptides	Proinflammatory signalling	[220]
TFAM transcription factor	Inflammatory response	[221]

### 6.3. Insulin Resistance

Insulin receptor (IR) is found in different brain areas, particularly in the hypothalamus, cortex and hippocampus [222]. It has been demonstrated that insulin enhances cognition and memory [223], modulates GABAergic activity in the cerebellum [224], protects against apoptosis [225], and regulates mitochondrial function [226,227].

Type 2 diabetes (T2D) is a pathological condition characterized by hyperglycemia, hyperinsulinemia and insulin resistance, and whose onset occurs in late adulthood and aging affecting millions of people worldwide. The pathophysiological relevance of T2D is also highlighted in brain aging, considering that T2D has been proposed as a risk factor for the development of AD. This relationship has ledto some authors terming AD as Type 3 Diabetes, since epidemiological studies have found insulin resistance to bea prodromal feature in AD patients [228,229]. Interestingly, oligomeric amyloid β-peptide (Aβ) binds to IR, impairing its function [230].

There is evidence linking insulin signaling and mitochondrial function. IR phosphorylation leads to brain mitochondrial biogenesis since it is able to activate the PI3K/AKT/mTOR pathway [231,232,233] (Figure 7). Moreover, the inhibition of the IR intracellular signaling has been shown to decrease mTOR signaling and PGC-1α levels [234,235], impairing mitochondrial biogenesis.

Classically, glycogen synthase kinase 3β (GSK3β) is inhibited by insulin signaling [236]. It is a neuronal protective mechanism induced by insulin, since GSK3β plays key roles in activating apoptosis by the induction of the mPTP, mitochondrial swelling, ΔΨm loss and CytC release [237,238]. Moreover, oxidative stress increases GSK3β activity and induces its translocation into mitochondria in a kinase activity-dependent manner, where the N-terminal domain of GSK3β may function as a mitochondrial targeting sequence [239]. These data suggest that GSK3β in mitochondria is a regulator of the mPTP opening, yielding to apoptosis. In fact, the inactivation of mitochondrial GSK3β is associated with suppression of mPTP opening [240,241]. Given that insulin signaling inhibits GSK3β, the lack of the proper intracellular activity mediated by PI3K/AKT suggests that GSK3β would play a deleterious role in neurons after the onset of insulin resistance.

The role of inflammaging [196] in insulin resistance can be relevant since pro-inflammatory cytokines activate IkB kinase (IKK) and c-Jun N-terminal kinase (JNK), which inhibit insulin signaling by phosphorylating the IR at serine-302 (pS302) and serine-307 (pS307), instead of its normal phosphorylating sites at tyrosine residues [242,243]. In addition, these cytokines also activate the suppressor cytokine signaling (SOCS) protein that binds and blocks the phosphorylated IR [244,245], contributing to mitochondrial dysfunction and triggering ROS production [244] (Table 3).

### 6.4. Alzheimer’s Disease

AD is characterized by the extracellular accumulation of Aβ misfolded into β-sheet in the brain [252,253,254]. Remarkably, nitro-oxidative stress is linked to AD as an etiogenic factor [137,255,256,257,258,259] (Table 4), which is tightly associated with mitochondrial dysfunction (Figure 8). In fact, there is increasing evidence that mitochondrial dysfunction is a key process in the onset and progression of AD [137,260,261]. Mitochondrial dysfunction and axonal degeneration are some of the first pathological characteristics of mild cognitive impairment (MCI), a clinical state that frequently precedes AD [262,263].

It has been demonstrated in samples from AD patients (at the early and late stages of AD) that complex I mitochondrial genes are downregulated. Increased expression of mtRNAs encoding complex III and IV proteins suggests a response to the increased energy demand [264,265,266]. Similar results have been found in an AD transgenic mouse model [267]. Interestingly, cytoplasmic hybrids (cybrid) derived from MCI and AD patients have higher ADP/ATP and lower nicotinamide adenine dinucleotide (NAD^+^/NADH) ratios [268].

With respect to mitochondrial biogenesis, postmortem studies with samples from AD patients showed an increase in the expression of Drp1, Fis1 and the mitochondrial matrix protein cyclophilin D (CypD). In contrast, expression of MFN1,2, OPA1 and Translocase of the OMM 40 (TOMM 40) were diminished. Aβ oligomers interact with Drp1, and this interaction increases as the disease progresses, leading to mitochondrial fragmentation, abnormal mitostasis and synaptic damage [269]. Moreover, presenilin 2 (PS2), part of the γ-secretase complex that produces Aβ by the cleavage at the C-terminus, has been shown to increase ER–mitochondria tethering by binding and sequestering MFN2 [123,270,271], therefore, affecting mitochondrial biogenesis.

Sirtuins (SIRT), in particular SIRT1, form a complex with PGC-1α transcriptional complex related with AD pathophysiology. It has been demonstrated that modest fasting reduces BACE1, the enzyme that cleavages the N-term of the amyloid precursor protein to produce Aβ, but increases PGC-1α expression and activity [272]. It is a unique, non-canonical mechanism by which SIRT1-PGC-1α activity induces transcriptional modulation in neurons in response to metabolic challenges reducing the Aβ load. Regarding mitochondria, SIRT1 overexpression protects against Aβ damages at the level of synaptic contacts, dendritic branching and mitochondrial morphology [273]. Furthermore, clusters of SIRT3 protein substrates have been identified in mitochondria, thereby implicating SIRT3 as a major mitochondrial SIRT involved in protecting mitochondrial integrity and energy metabolism under stress conditions. In fact, theloss of this protein has been found to accelerate brain neurodegeneration challenged in excitotoxicity [274,275]. 

Regarding the neuroprotective Wnt signaling, the activation of the β-catenin-independent pathway by Wnt5a ligand results in the modulation of mitostasis, preventing the changes induced by Aβ in mitochondrial fission–fusion dynamics and modulating the Aβ-induced increase in antiapoptotic Bcl-2 mitochondrial exposure [276]. The activation of the β-catenin-dependent pathway by the Wnt3a ligand has been demonstrated to inhibit the mPTP opening induced by Aβ [277]. On the other hand, the Wnt-induced signaling protein 3 (WISP3/CCN6) has also been related with the appropriate functioning of the mitochondria [278]. Indeed, there are increased mitochondrial ROS and Ca^2+^ levels in WISP3-depleted cells [279]. These findings suggest that the activation of Wnt pathways prevents mPTP opening and the mechanisms implied in this activity involve the inhibition of mitochondrial GSK3β and/or the modulation of mitochondrial hexokinase II (HKII) levels and activity [280].

Aβ also induces an increase in intracellular Ca^2+^ by the interaction with L/P/Q-types VGCC [281], NMDAR [282], RyR [283,284] and indirectly by other mechanisms [285,286]. Interestingly, some of the familial forms of AD are due to PS1 mutations which do notaffect Aβ production but rather increase the Ca^2+^ leak from the ER to the cytoplasm, highlighting the role of Ca^2+^ in neuronal apoptosis [287]. Moreover, it has been demonstrated that Aβ increases ER–mitochondria coupling and the Ca^2+^ transfer from the ER to mitochondria [127,288]. This increase in intracellular Ca^2+^ leads to mitochondrial dysfunction and initiation of apoptosis. Likely, the loss of Ca^2+^ regulation by the action of Aβ is one of the main causes of the increased production of free radicals and peroxynitrite formation that induces the nitrotyrosination of proteins in AD [168,169,289,290,291].

On the other hand, mitochondrial analysis from AD patient samples showed a significant decrease in mitochondria number in vulnerable neurons that also showed oxidative damage and nitrotyrosination [292,293]. These neurons contained increased levels of mitochondrial degradation products, suggesting either an increased turnover of mitochondria via autophagy or reduced proteolytic turnover leading to accumulation of mtDNA and mitochondrial proteins [292,293]. Interestingly, base excision repair (BER) is one of the most active DNA repair pathways in cells, correcting chromosomal and mtDNA damage from oxidation. It is a process that occurs under conditions of oxidative stress, which may lead to double-strand breaks and induce apoptosis [294]. BER deficiency and polymorphisms have been associated with the development of neurodegenerative diseases, such as AD and ALS, due to the reduction in DNA repairment [295,296].

**Table 4 biomolecules-11-01012-t004:** Proteins involved in mitochondrial dysfunction in Alzheimer’s disease.

Molecule	Function	Ref.
Cytochrome c oxidase and mitochondrial F1F0-ATPase (ATP synthase)	MRC	[297]
ERK	Oxidative stress-mediated ERK signal	[298]
Amyloid beta (Aβ) binding alcohol dehydrogenase (ABAD)	Promotes Aβ-mediated mitochondrial a dysfunction	[299]
mtNADH dehydrogenase 1 (mt-Nd1)	MRC	[300]
OPA1, Mfn1, Mfn2, DRP1 and Fis1	Mitochondrial fusion	[269]
MT-ND2 and MT-ND5	MRC	[301]

### 6.5. Parkinson’s Disease

PD is a progressive neurodegenerative disease characterized by the pathological loss of dopaminergic neurons in the substantia nigra pars compacta (SNPC) [302]. Mitochondrial dysfunction has been associated with the pathogenesis of PD since drug abusers were accidentally exposed to 1-methyl4-phenyl-1,2,3,4-tetrahydropyridine (MPTP), which induced acute parkinsonian syndrome due to a metabolite of MPTP inhibiting complex I in the MRC resulting in neuronal death [303]. Since this discovery, further support for the link between mitochondrial defects and PD has been revealed due to genetic factors inducing familial PD such as Parkin, PTEN-induced kinase 1 (PINK1), DJ-1, alpha-synuclein, leucine-rich repeat kinase 2 (LRRK2) and coiled-coil-helix-coiled-coil-helix domain-containing 2 (CHCHD2) (Table 5), which are involved in the function and turnover of mitochondria, impaired mitochondrial dynamics and function in the pathogenesis of PD [304,305] (Figure 9).

The ubiquitin ligase, Parkin and the mitochondrial kinase PINK1 control the autophagic clearance of defective mitochondria. DJ-1 maintains mitochondrial function in response to oxidative stress [306,307]. Furthermore, PGC-1α, the master regulator of mitochondrial biogenesis and oxidative stress resistance, has been shown to be blocked in familial PD with Parkin mutation [308]. Consistently, the activation of PGC-1α has showed a protective role in dopaminergic neurons when α-synuclein is mutated [306,309].

In PD there is an increase in intracellular Ca^2+^ caused by L- and N-type VGCC, which can cause saturation of mitochondrial buffering capacity and eventually apoptosis [310]. Ca^2+^ overload may force the opening of mPTP, leading to retrograde electron flux through the MRC, resulting in increased ROS production, release of Cyt C, and activation of apoptosis [311]. As commented before, MAM allows Ca^2+^ to be funneled from the ER to the mitochondria via the complex formed by IP_3_R, VDAC and GRP75, and the activity of this complex is disrupted by α-synuclein [312]. Mutations in α-synuclein also reduce the binding to MAM and increase mitochondrial fragmentation, demonstrating its regulatory role in mitochondrial morphology [313]. Moreover, α-synuclein has been found to influence mitochondrial structure and function, inhibiting complex I from the MRC [314].

Parkin is recruited to damaged or dysfunctional mitochondria in the earlystages, activated by PINK1, resulting in the ubiquitination of OMM proteins and subsequent proteasome degradation. Parkin-mediated mitophagy was found in the distal axons of rodent neurons and in age-related dopaminergic neurodegeneration in Parkin knockout mice with defective mitochondrial DNA replication [315,316].

LRRK2 contributes to mitophagy via its interference with mitochondrial trafficking, as its mutant form in a model of induced pluripotent stem cell-derived neurons impairs proteasome degradation of Miro, the OMM protein. This tethers mitochondria to microtubule motor proteins, resulting in mitophagy due to the disrupted interaction between LRRK2 and Miro [317].

Finally, it has been reported that CHCHD2 or Mitochondria Nuclear Retrograde Regulator 1 (MNRR1) is bound to the mitochondrial complex IV, and its reduced expression resulted in a decrease in mitochondrial complex IV activity, increasing ROS production and mitochondrial fragmentation. This MNRR1-mediated stress response may provide an important survival mechanism for cells under conditions of oxidative or hypoxic stress, both in the acute phase by altering mitochondrial oxygen utilization [318].

**Table 5 biomolecules-11-01012-t005:** Proteins involved in mitochondrial dysfunction in Parkinson disease.

Molecule	Function	Ref.
MITOL/MARCH5	Recruitment to dysfunctional mitochondria	[319]
p97 (AAA + ATPase)	Degradation of ubiquitinated Mfn1	[320]
Tim23 complex	Translocation of the positively charged MTS across the IMM	[321]
HTRA2/Omi protease	Regulated by PINK 1	[322]
TRAP1	Regulated by PINK 1 in IMS	[323]
complex I subunit NDUFA10	Regulated by PINK 1 in IMM	[324]
PINK1/Parkin	Mutated PINK1 activates Parkin causing mitophagy	[325]
PGC-1α	Blocked by mutated Parkin	[308]
L- and N-type VGCC	Increase of cytosolic calcium	[310]
α-synuclein	MAMs disruption and complex I from MRC inhibition when it is mutated	[312,314]
DJ-1	Oxidative stress sensor and neuroprotective	[326]
LRRK2	Its mutation reduces mitophagy through its disrupted interaction with Miro and a blockade of PINK1/Parkin-dependent mitophagy.	[317,327]
CHCHD2	Regulation of complex IV activity	[318]

### 6.6. Amyotrophic Lateral Sclerosis

ALS is a neurodegenerative disease that compromises lower and upper motor neurons. The causes are unknown but 5% of the cases are due to genetic background. These genetic causes include hexanucleotide repeated insertions in C9ORF72 [328,329], or mutations mostly in SOD1 [330], fused in sarcoma (FUS) [331,332] and transactive response DNA binding protein (TARDBP) [333,334] genes (Figure 10). TARDBP gene encodes for TDP-43, an RNA-binding protein, whose aggregates inside the motor neurons are the most significant pathological finding in ALS [335] (Table 6). TDP-43 accumulation also compromises mitostasis, affecting fission and fusion kinetics, mitochondrial transport, bioenergetics, and mitochondrial quality control [336].

A mutation in the mitochondrial protein coiled-coil-helix-coiled-coil-helix domain-containing 10 (CHCHD10), CHCHD10^S59L^, has also been associated with frontotemporal dementia and ALS [337]. It drove progressive motor deficits, myopathy, cardiomyopathy and accelerated mortality in a mutated mice model [338] and disrupted OPA1-mitofilin complexes in transfected cells and in vivo mice [339]. Evidence from SOD1^G93A^ rat models (substitution of glycine to alanine at position 93) has demonstrated that misfolded SOD1 is able to increase toxicity due to a toxic gain-of-function role, and exists in different conformers dependent on the type of misfolding [340], which deposit on mitochondrial subpopulations on the OMM, IMM, intermembrane space and matrix [341,342]. Consistently, a model of human SOD1^G93A^ transgenic pigs, which displayed ALS-like symptoms, showed impaired mitostasis [343]. Interestingly, it has been suggested that a defect in mitophagy may contribute to the development of ALS, as fewer phagosomes are found than in controls using mice models [344]. This defect could be explained by a mutant SOD1-mediated optineurin sequestration [345]. The mitophagy-associated proteins Parkin and PINK1 have also been found to be downregulated in ALS mice models [346].

Impairment of mitochondrial function can result in excess production of ROS, and due to the proximity to the ROS production site, mtDNA is highly prone to damage. Furthermore, mtDNA is not associated with histones and chromatin-associated proteins for protection and is highly vulnerable to damage [347]. On the other hand, in both sporadic and familial cases of ALS it has been proposed that the excitotoxicity by glutamate could play a key role [348]. This might be due to an increased Ca^2+^ influx that would affect mitochondrial function and would initiate apoptosis.

**Table 6 biomolecules-11-01012-t006:** Proteins involved in mitochondrial dysfunction in ALS.

Molecule	Function	Ref.
SOD-1	Its aggregation impairs proper mitophagy by optineurin sequestration, and axonal transport of mitochondria	[345]
TDP-43	Involved in interaction with mitochondrial proteins	[349]
TIM22	TDP-43 gains access to the mitochondrial matrix via the mitochondrial import inner membrane	[350]
CHCHD10	Involved in mitochondrial defects	[337]
Dbr1	suppresses TDP-43 toxicity	[351]
Ataxin 2	Increased risk for ALS	[352]
GSK-3β	disrupt the VAPB PTPIP51 interaction and ER–mitochondria associations	[353]
Alsin	Increased susceptibility to oxidative stress due to a defect in Rab5 relocation to mitochondria	[354]

### 6.7. Huntington’s Disease

HD is a neurodegenerative disease caused by the repetition of CAG in the huntingtin gene, which leads to huntingtin protein (HTT) aggregation (Table 7), inducing neuronal death, mainly in the corpus striatum but also in systemic cells [355]. In HD there is an increase in the expression of the functional subunits of NMDAR due to HTT [356], where NO could be playing a translational role [357,358]. This increase produces a higher and maintained intracellular Ca^2+^ concentration, affecting mitochondrial function [359] and triggering apoptosis [360] (Figure 11).

HTT is also associated with increased oxidative stress, since the aggregation of the N-terminus fragments of the mutant HTT contributes to an increased generation of ROS [361]. On the other hand, HTT inclusion bodies generate iron-dependent oxidative stress [362]. One of the pathological effects of the increased ROS production is the oxidation of the mtDNA [363], which impairs mitochondrial biogenesis as it has been reported in HD [364,365]. Another factor that contributes to impair mitochondrial biogenesis in HD is the direct interaction of mutated HTT with Drp1 [366].

**Table 7 biomolecules-11-01012-t007:** Proteins involved in mitochondrial dysfunction in Huntington disease.

Molecule	Function	Ref.
Huntingtin	Its aggregation increases mitochondrial calcium concentration and ROS production	[356]
Ubiquitin-like protein 1 (NUB1)	Suppressor of mutant Huntington toxicity via enhanced protein clearance	[367]
Bcl-2/adenovirus E1B 19-kDa interacting protein 3 (BNip3)	Evidence of an abnormal activation in cells expressing mutant Huntingtin.	[368]
p53	Regulates the levels of huntingtin gene expression	[369]
BDNF	Enhancing BDNF vesicular transport along microtubules	[370]
Alpha-tubulin deacetylase (HDAC)	Pharmacological inhibition of the HDAC6 deacetylase activity increased acetylated alpha-tubulin levels, and induced mitochondrial motility and fusion in striatal neurons	[371]
PGC1a	Transcriptional repression of PGC-1alpha by mutant huntingtin	[372]
Drp1	Its interaction with mutant HTT impairs mitochondrial biogenesis, causing defective mitochondria axonal transport and synaptic degeneration	[366]

## 7. Conclusions

Aging and neurodegenerative processes are not the consequences of a unique signaling pathway alteration but of multifactorial events, including dysfunction of mitochondria that yield to failures in ATP production, increased ROS production and alterations in Ca^2+^ homeostasis. Considering mitochondria as a therapeutictarget would produce an improvement in the maintenance of the optimum neuronal function. These therapeutic approaches would include the regulation of PGC-1α and calcium buffering systems as well as free radicals’generation, addressing the source of free radicals more than the use of antioxidants that have not demonstrated effective results to date.

## Figures and Tables

**Figure 1 biomolecules-11-01012-f001:**
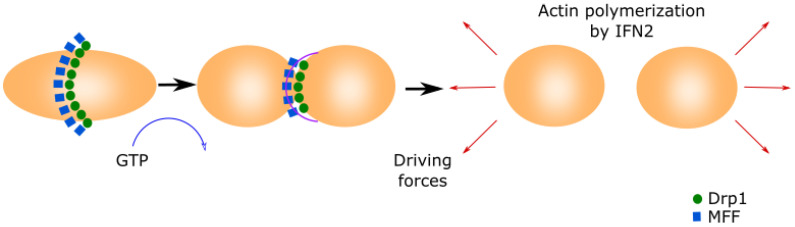
Mitochondrial fission.This process is initiated by MFF forming a ring in the middle of the primary organelle to be divided. Drp1 GTPase translocates to the mitochondria and binds to MFF, inducing a conformational change by the hydrolysis of GTP. This conformational change generates the ring contraction, which starts mitochondrial fission. The reorganization of the cytoskeleton generates the driving forces that separate both new mitochondria.

**Figure 2 biomolecules-11-01012-f002:**
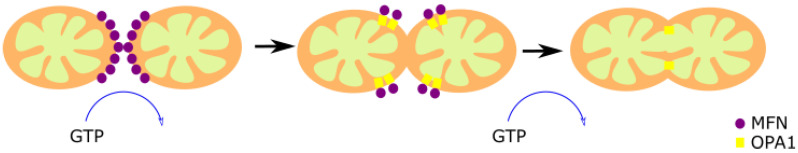
Mitochondrial fusion. The fusion of two preexistent mitochondria is a protective mechanism based on the fusion of the OMM and IMM. The OMM fusion is mediated by MFN1 and MFN2, and the fusion of the IMM is mediated by OPA1. First, MFN1 and MFN2 GTPase activity induce the formation of homo- or hetero-oligomeric complexes in the fusion spot. Then, OPA1-mediated GTP hydrolysis promotes the membrane fusion.

**Figure 3 biomolecules-11-01012-f003:**
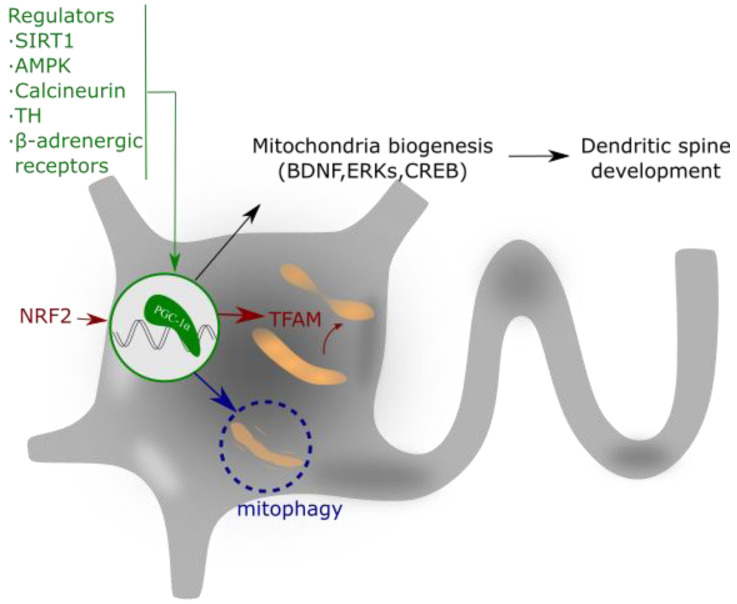
PGC-1α regulates mitochondrial biogenesis and spine growth. Its activity is based on the interaction with NRF that activates to TFAM. This complex regulates mitochondrial gene expression and addresses the requirements of neuronal energy when needed for dendrite activity or growth. PGC-1α expression is regulated by different intracellular signaling pathways, as explained in the text.

**Figure 4 biomolecules-11-01012-f004:**
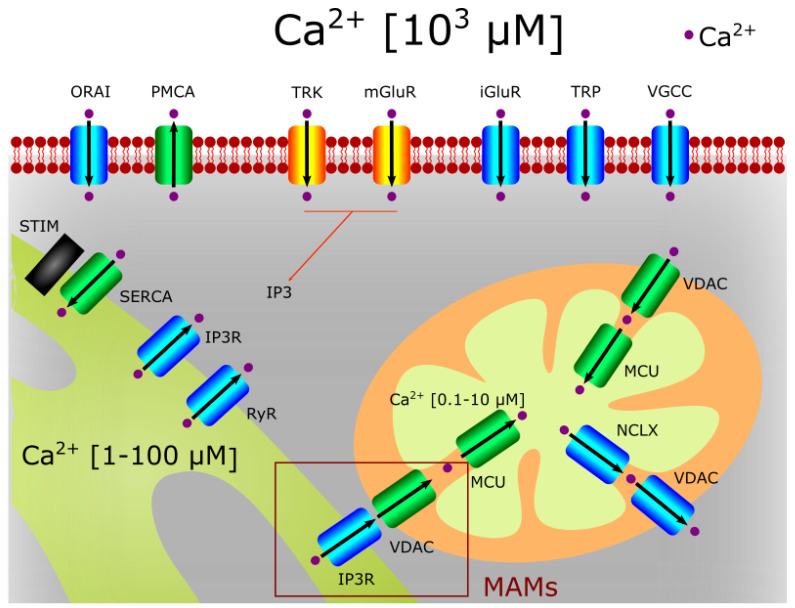
Ca^2+^ homeostasis in neurons. The entrance of extracellular Ca^2+^ into neurons is due to the direct activation of neurotransmitter and neurotrophin receptors (such as NMDAR and TRK) and cationic channels (such as TRPs or voltage activated Ca^2+^ channels). The VDAC channel can import or export Ca^2+^ in the mitochondria when associated to MCU or NCLX, respectively. IP3R and VDAC channels found in MAMs can also transport Ca^2+^ from the ER to the mitochondria. The Ca^2+^ extrusion to the extracellular medium is mediated by ATPases (i.e., PMCA).

**Figure 5 biomolecules-11-01012-f005:**
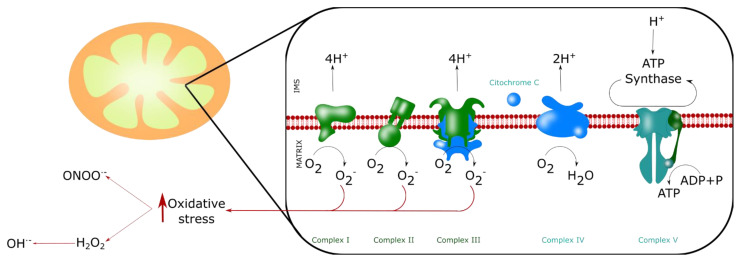
ROS production by mitochondria. Mitochondria produce ROS, mainly superoxide anion O_2_·^−^ by the MRC (at the complexes I, II and III), and indirectly hydroxyl radical (OH·), hydrogen peroxide (H_2_O_2_) andperoxynitrite (ONOO^−^). These four molecules are the main reactive agents inducing nitro-oxidative stress.

**Figure 6 biomolecules-11-01012-f006:**
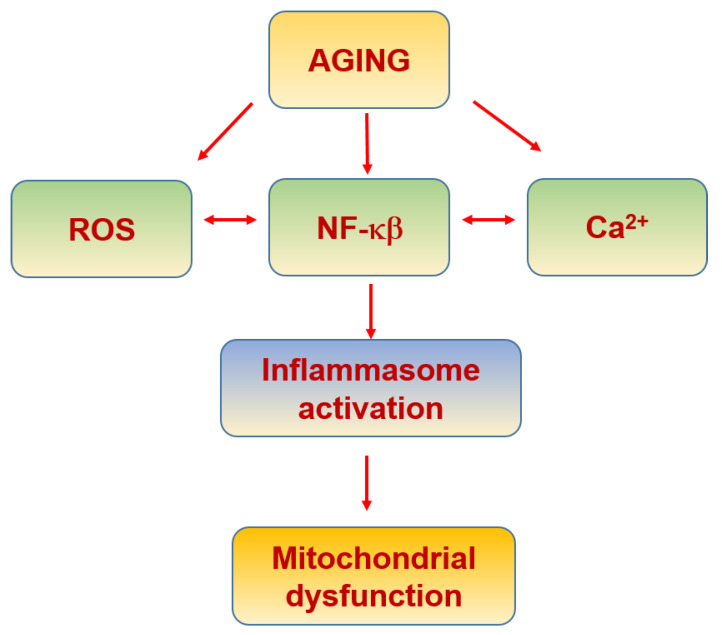
Inflammaging and mitochondria.

**Figure 7 biomolecules-11-01012-f007:**
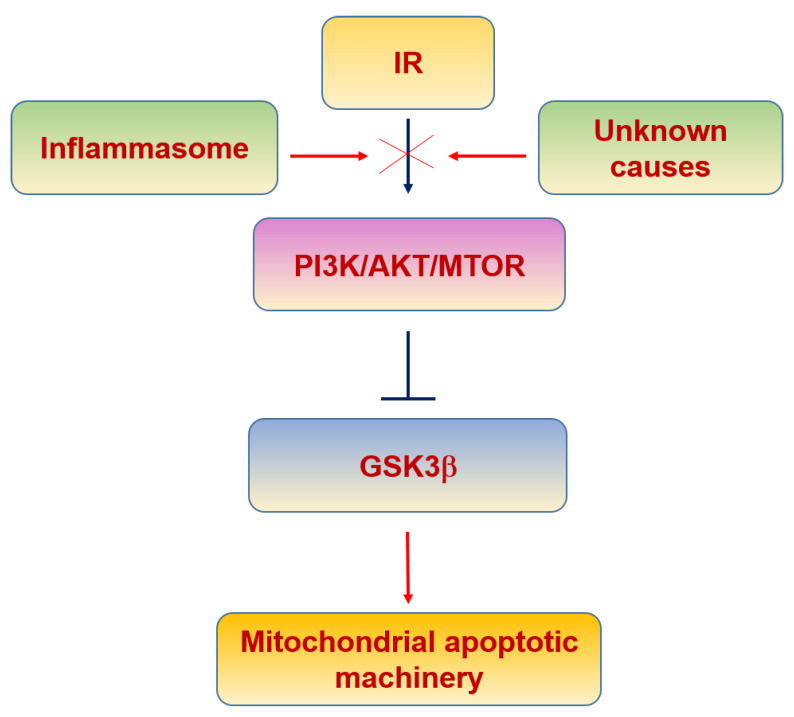
Insulin resistance and mitochondria.

**Figure 8 biomolecules-11-01012-f008:**
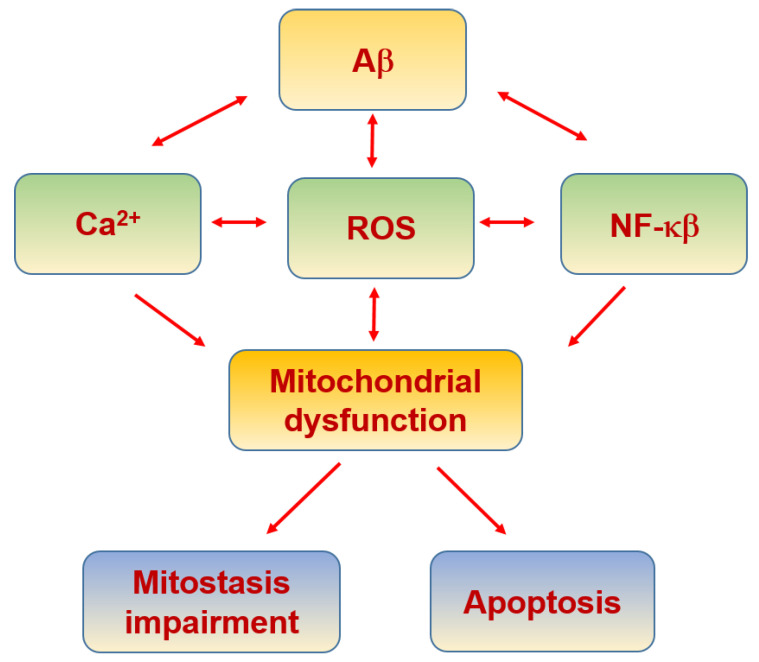
Alzheimer’s disease and mitochondria.

**Figure 9 biomolecules-11-01012-f009:**
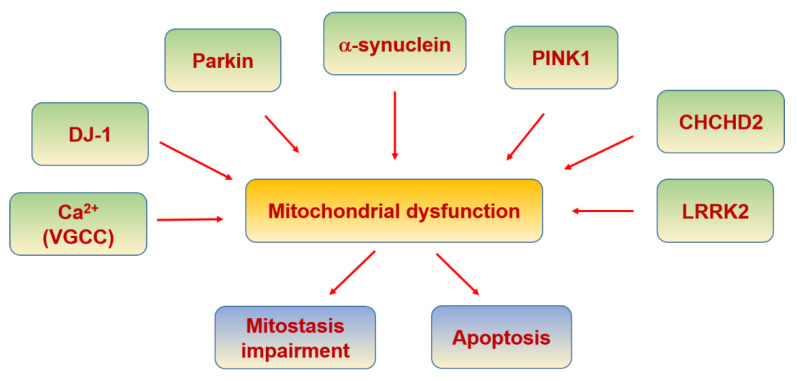
Parkinson’s disease and mitochondria.

**Figure 10 biomolecules-11-01012-f010:**
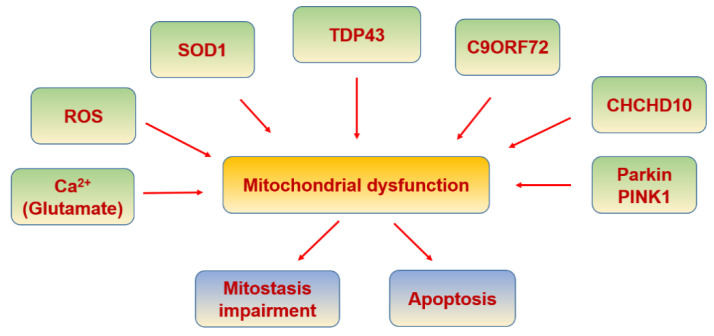
Amyotrophic lateral sclerosis and mitochondria.

**Figure 11 biomolecules-11-01012-f011:**
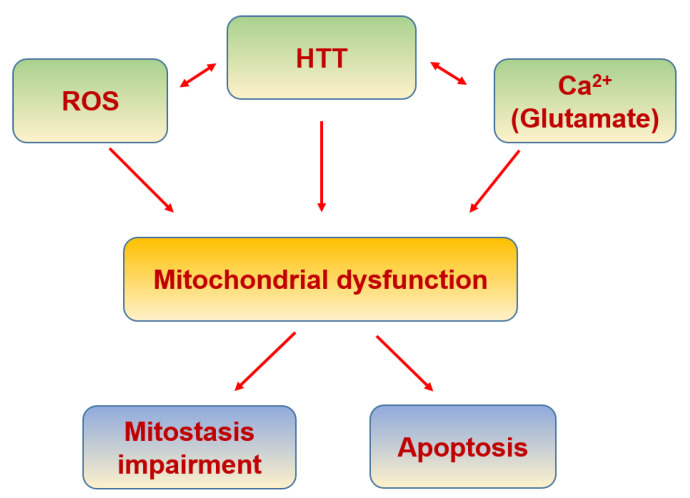
Huntington’s disease and mitochondria.

**Table 3 biomolecules-11-01012-t003:** Proteins involved in mitochondrial dysfunction in insulin resistance.

Molecule	Function	Ref.
NADH:O2 oxidoreductase	Oxidoreductase activity	[246]
NADH dehydrogenase NDUFB8, succinate dehydrogenase SDHB and COX4I1	Encoding a complex I accessory subunit to reducemitochondrial activity	[247]
Nitric oxide (NO)	Mitochondrial biogenesis at the transcriptional level through activation of cGMP and PGC1α	[161]
Hydrogen sulfide (H2S)	Preserves mitochondrial function via increased AKT phosphorylation, increased nuclear localization of NRF1/2, and increased mitochondrial biogenesis	[248]
Estrogen 17beta-estradiol	Transcription of nuclear respiratory factor-1 and increases mitocondrial biogenesis.	[249]
FOXO 1	Mitochondrial function	[250]
PPARγ	Glucose metabolism and regulation of inflammatory response	[251]

## Data Availability

Not applicable.

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
