# Peer review of "Mitostasis, Calcium and Free Radicals in Health, Aging and Neurodegeneration"

_biomolecules, 2021, doi:10.3390/biom11071012_

Round 1

Reviewer 1 Report

The authors work is based on two assumptions - mitochondrial uptake of calcium is large and mitochondrial associate membranes allow calcium signaling crosstalk between ER and mitochondria.  There have been many papers suggesting this, but when evaluated critically the link is tenuous.  There is need for a thoughtful critical analysis rather than jumping on the the bandwagon.

The manuscript hops from subject to subject.  It might be helpful to have better organization.  Perhaps a statement early on describing the topics to be covered and how they fit together.  It tries to link everything to MAM’s.  The focus seems to be the role of mitochondrial in physiology and pathophysiological process.  Sometimes in the literature MAMs have been associated with these phenomena.   However, the causality and functional consequences are still an area of active study.    The authors need to be more critical of the evidence and presenting it.

Line 49 – The authors state that mitochondria are key regulators of calcium homeostasis.  What is the evidence behind this assumption?  Recent studies with genetically encoded dyes have shown that mitochondrial calcium uptake is actually quite small.  Other studies with other dyes have shown otherwise, but it is not clear that the dyes were actually making it into the mitochondrial matrix.  This needs to be discussed.

Line 51 – The work suggesting that there is mitochondrial-ER crosstalk is speculative.  There is not clear data showing that this is the case.  The authors present this as a fact.  This assumption need to be discussed more fully.

There are numerous typographical errors and places where wording can be improved.  Some examples are as follows:

Line 396 – “Imflammaging”?  maybe it should be “Inflammation”

Line 437 – two missing characters

Line 471 – “Attending to…”  - the authors might want to reword.

Line 475 – Nicotinamide should note be capitalized.

Line 484 – missing something before secretase

Line 517 – missing beta from Abeta

Line 522 – extra space

Author Response

# Reviewer 1 #

 “The authors work is based on two assumptions - mitochondrial uptake of calcium is large and mitochondrial associate membranes allow calcium signaling crosstalk between ER and mitochondria.  There have been many papers suggesting this, but when evaluated critically the link is tenuous.  There is need for a thoughtful critical analysis rather than jumping on the the bandwagon.”

Dear reviewer, we understand your criticism because it is a wide review on different aspect affecting mitochondria. Please believe us that as researchers the issues we have addressed are the ones that really concern us in our basic research and this review is an exercise that comes from our experience in mitochondria, calcium, nitro-oxidative stress, mitostatis and neurodegenerative diseases to clarify certain aspects that in the international literature are contradictory and that we have reflected throughout the review.

“The manuscript hops from subject to subject.  It might be helpful to have better organization.  Perhaps a statement early on describing the topics to be covered and how they fit together.  It tries to link everything to MAM’s.  The focus seems to be the role of mitochondrial in physiology and pathophysiological process.  Sometimes in the literature MAMs have been associated with these phenomena.   However, the causality and functional consequences are still an area of active study. The authors need to be more critical of the evidence and presenting it.”

It is well known that the mitochondrial calcium uptake depends on the release of calcium from the endoplasmic reticulum in MAMS. Malfunction of IP3R, the channel that allows calcium exit from the ER that localize in MAMs, is associated with unfolded protein response (UPR) and cell dead, which is a common event to all the neurodegenerative diseases. For this reason we have included MAMs in the review but it is just a section in a review with more than 20 sections.

We agree with the reviewer that this is an area of active study. Nevertheless, we think that it is essential to highlight in the review the structures that allow the ER-mitochondrial calcium crosstalk. Following reviewer’s comment we have modified or eliminated the sentences regarding MAMs function that were overstated or too speculative.

“Line 49 – The authors state that mitochondria are key regulators of calcium homeostasis.  What is the evidence behind this assumption?  Recent studies with genetically encoded dyes have shown that mitochondrial calcium uptake is actually quite small.  Other studies with other dyes have shown otherwise, but it is not clear that the dyes were actually making it into the mitochondrial matrix.  This needs to be discussed.”

Dear reviewer, the use of genetically-encoded calcium indicators has revealed the existence of transient changes in the mitochondrial calcium levels that reach from low micromolar to submillimolar concentrations in a fraction of mitochondria from stimulated cells (Arnaudeau S., Kelley W.L., Walsh J.V., Demaurex N. Mitochondria recycle Ca2+ to the endoplasmic reticulum and prevent the depletion of neighboring endoplasmic reticulum regions. J. Biol. Chem. 2001;276:29430–29439. doi: 10.1074/jbc.M103274200; Wüst R.C.I., Helmes M., Martin J.L., van der Wardt T.J.T., Musters R.J.P., van der Velden J., Stienen G.J.M. Rapid frequency-dependent changes in free mitochondrial calcium concentration in rat cardiac myocytes. J. Physiol. (Lond.) 2017;595:2001–2019. doi: 10.1113/JP273589). These changes in calcium fit perfectly within the physiological and pathological windows of mitochondrial calcium levels mentioned through the manuscript (1-100 µM). On the other hand, there are strong evidences showing that alterations in gatekeepers of MCU complex like EMRE, MICU1 and or m-AAA protease impact neuronal survival by an excess of calcium accumulation. In the same direction, the overexpression of MCU causes neuronal dead. Following reviewer’s comment we have modified or eliminated the sentences regarding mitochondrial calcium that were overstated or too speculative. We have included the range [1-100 µM] in the mitochondria in the new figure 4.

“Line 51 – The work suggesting that there is mitochondrial-ER crosstalk is speculative.  There is not clear data showing that this is the case.  The authors present this as a fact.  This assumption need to be discussed more fully.”

We have modified the sentences along the manuscript that indicate this kind of relationship and the word crosstalk has been eliminated in the whole manuscript.

“Line 396 – “Inflammaging”?  maybe it should be “Inflammation”

            Inflammaging is a new term to define a low degree of inflammatory response in different cell types associated to aging, which is not a constitutive disease per se but it is inducing an insidious damage in brain and systemic organs. Please have a look in Pub Medline and you will find more than 1000 references with this term. It is very interesting and useful term to be applied in aging pathophysiology.

“Line 437 – two missing characters”

The typos have been corrected.

“Line 471 – “Attending to…”  - the authors might want to reword.”

Thank you very much. It has been corrected.

“Line 475 – Nicotinamide should not be capitalized.; Line 484 – missing something before secretase; Line 517 – missing beta from Abeta; Line 522 – extra space”

The typos have been corrected.

Reviewer 2 Report

This article reviews the movement and regulatory effects of calcium in and between endoplasmic reticulum and mitochondria. The focus is on neurons and neurodegenerative diseases. This is a wide subject, but this review maintains focus relatively well. Overall this article is useful to researchers in the field.

Minor comments:

-Line 53: dyshemostasis – dyshomeostasis

-Line 128, line 133-140: MNF1,2 – MFN1,2 

-Line 143: optics – optic 

-Figure 3 and/or the legend could benefit from some clarification. The process of mitophagy is in the figure but not commented in the legend. TFAM is hardly the only gene that PGC1 alpha regulates? The image implies that mitochondrial biogenesis induced by BDNF, ERKs and CREB is independent of PGC1alpha, is this the case? 

-Figure 4: It is somewhat misleading to write Ca2+ in big font in the cytoplasm as that implies high basal concentrations of Ca2+ in the cytosol. It would be interesting to see approximate basal Ca2+ concentrations for the different compartments (ER, matrix, intermembrane space, cytosol, extracellular). It would also be of interest to see which of the Ca2+ currents are ATP-dependent.

-line 533: It would be good also to mention C9ORF72 mutation as a genetic cause of ALS, as this is more common than SOD1 mutations in many countries. 

-It would be relevant to briefly discuss CHCHD10, which is a mitochondrial intermembrane space protein, and where mutations predispose to ALS and other neurodegenerative diseases. 

Author Response

# Reviewer 2 #

 “This article reviews the movement and regulatory effects of calcium in and between endoplasmic reticulum and mitochondria. The focus is on neurons and neurodegenerative diseases. This is a wide subject, but this review maintains focus relatively well. Overall this article is useful to researchers in the field.”

Dear reviewer, thank you very much for your positive comment on our work.

“-Line 53: dyshemostasis – dyshomeostasis; Line 128, line 133-140: MNF1,2 – MFN1,2; Line 143: optics – optic”

The typos have been corrected.

“-Figure 3 and/or the legend could benefit from some clarification. The process of mitophagy is in the figure but not commented in the legend. TFAM is hardly the only gene that PGC1 alpha regulates? The image implies that mitochondrial biogenesis induced by BDNF, ERKs and CREB is independent of PGC1alpha, is this the case?”

We have modified completely the figure 3 and its legend to clarify the concerns raised by the reviewer.

“-Figure 4: It is somewhat misleading to write Ca2+ in big font in the cytoplasm as that implies high basal concentrations of Ca2+ in the cytosol. It would be interesting to see approximate basal Ca2+ concentrations for the different compartments (ER, matrix, intermembrane space, cytosol, extracellular). It would also be of interest to see which of the Ca2+ currents are ATP-dependent.”

We have modified the figure 4 and its legend according reviewer’s comments.

“-line 533: It would be good also to mention C9ORF72 mutation as a genetic cause of ALS, as this is more common than SOD1 mutations in many countries. -It would be relevant to briefly discuss CHCHD10, which is a mitochondrial intermembrane space protein, and where mutations predispose to ALS and other neurodegenerative diseases.”

 Thank you very much. We have included both mutations in the Neurodegeneration Section of the manuscript.

Reviewer 3 Report

Please see the attached review report.

Author Response

# Reviewer 3 #

 “In this review article, the authors have summarised the importance of mitochondrial homeostasis in maintaining neuronal function and regulation of this

homeostasis is important in maintaining neuronal viability, protecting against aging and neurodegeneration. They have done a detailed comprehensive summary of mitochondrial homeostasis, role of mitochondria in Calcium homeostasis, Reactive oxygen species production by mitochondria, regulation of mitochondrial function and homeostasis by Nitric oxide and role of mitochondria in aging and neurodegeneration. This review provides detailed citations of articles which have concluded that Mitochondria play key roles in ATP supply, calcium homeostasis, redox balance control and apoptosis. These functional roles of mictochondria in neurons are fundamental for neurotransmission and aids in synaptic plasticity. The authors have defined the role of mitostasis, a process that involved mitochondrial transport, anchoring, fusion and fission, processes regulated by different signaling pathways but mainly by the peroxisome proliferator-activated receptor-g coactivator-1a (PGC-1a). PGC-1alpha favors Ca2+ homeostasis, reduces oxidative stress, modulates inflammatory processes and mobilizes mitochondria. The review article also states that mitochondria are tightly connected to the endoplasmic reticulum (ER) through specialized structures of the ER termed mitochondriaassociated membranes (MAMs), which facilitate the crosstalk between them mainly to aim Ca2+ buffering. Alterations in mitochondrial activity enhances reactive oxygen species (ROS) production disturbing the physiological metabolism and causing cell damage. On the other hand, cytosolic Ca2+ overload leads to an increase in mitochondrial Ca2+, resulting in mitochondrial dysfunction and the induction of mitochondrial permeability transition pore (mPTP) opening, leading to mitochondrial swelling and cell death through apoptosis as demonstrated in several

neuropathologies.

This review is within the aims and scope of biomolecules as it is covering the role of genes / biomolecules involved in maintaining mitochondrial homeostasis, trafficking, ros production, role of NO in mitostasis, mitochondrial dysfunction and the overall implication of all these in maintaining neuronal function, synaptic

transmission and their failure in neuronal degeneration.

Overall Recommendation: Accept after Revisions: The paper is in principle

accepted after minor revisions based on the reviewer’s comments.”

Dear reviewer, thank you very much for your exhaustive and positive comment on our work.

“However, there are few points that need to be addressed in the paper:

  1. I would like to have a schematic of mitochondrial genes and other effectors in physiological aging, imflammaging, Insulin resistance and neurodegenerative disease like AD, PD, Huntington disease and ALS – this could be one diagram or 3 diagrams.”

Following reviewer’s recommendation we have include schemes as figures for the different pathological states revised in our work to clarify the role of mitochondria dysfunction in each particular state.

  1. A table of all the mitochondrial genes and the mitochondrial processes affected in each of these disease should also be listed.

            As the rewiever has suggested, we have included tables with the main genes/proteins involved in aging and neurodegenerative diseases.

Round 2

Reviewer 1 Report

The authors have addressed my concerns.